# Development of Dog Vaccination Strategies to Maintain Herd Immunity against Rabies

**DOI:** 10.3390/v14040830

**Published:** 2022-04-16

**Authors:** Ahmed Lugelo, Katie Hampson, Elaine A. Ferguson, Anna Czupryna, Machunde Bigambo, Christian Tetteh Duamor, Rudovick Kazwala, Paul C. D. Johnson, Felix Lankester

**Affiliations:** 1Environmental Health and Ecological Sciences Department, Ifakara Health Institute, Dar es Salaam P.O. Box 78373, Tanzania; 2Boyd Orr Centre for Population and Ecosystem Health, Institute of Biodiversity, Animal Health and Comparative Medicine, University of Glasgow, Graham Kerr Building, Glasgow G12 8QQ, UK; katie.hampson@glasgow.ac.uk (K.H.); elaine.ferguson@glasgow.ac.uk (E.A.F.); anna.czupryna@glasgow.ac.uk (A.C.); paul.johnson@glasgow.ac.uk (P.C.D.J.); 3Department of Veterinary Medicine and Public Health, Sokoine University of Agriculture, Morogoro P.O. Box 3105, Tanzania; kazwala@gmail.com; 4Global Animal Health Tanzania, Arusha 1642, Tanzania; bigambochunde@gmail.com (M.B.); felix.lankester@wsu.edu (F.L.); 5Department of Global Health, Nelson Mandela African Institution of Science and Technology, Arusha P.O. Box 447, Tanzania; ctetteh@ihi.or.tz; 6Paul G. Allen School for Global Health, College of Veterinary Medicine, Washington State University, Pullman, WA 99164, USA

**Keywords:** rabies, vaccination strategy, herd immunity, decentralized continuous vaccination, mass dog vaccination

## Abstract

Human rabies can be prevented through mass dog vaccination campaigns; however, in rabies endemic countries, pulsed central point campaigns do not always achieve the recommended coverage of 70%. This study describes the development of a novel approach to sustain high coverage based on decentralized and continuous vaccination delivery. A rabies vaccination campaign was conducted across 12 wards in the Mara region, Tanzania to test this approach. Household surveys were used to obtain data on vaccination coverage as well as factors influencing dog vaccination. A total 17,571 dogs were vaccinated, 2654 using routine central point delivery and 14,917 dogs using one of three strategies of decentralized continuous vaccination. One month after the first vaccination campaign, coverage in areas receiving decentralized vaccinations was higher (64.1, 95% Confidence Intervals (CIs) 62.1–66%) than in areas receiving pulsed vaccinations (35.9%, 95% CIs 32.6–39.5%). Follow-up surveys 10 months later showed that vaccination coverage in areas receiving decentralized vaccinations remained on average over 60% (60.7%, 95% CIs 58.5–62.8%) and much higher than in villages receiving pulsed vaccinations where coverage was on average 32.1% (95% CIs 28.8–35.6%). We conclude that decentralized continuous dog vaccination strategies have the potential to improve vaccination coverage and maintain herd immunity against rabies.

## 1. Introduction

Rabies remains a major public health problem causing over 60,000 human deaths every year worldwide. More than 95% of these fatalities occur in Africa and Asia, where canine rabies is endemic [1]. In addition to causing human mortality, anxiety and economic burden to those who are exposed, rabies further impacts livelihoods through losses of livestock [2]. Domestic dogs are the principle reservoir of the virus in Africa and are responsible for causing more than 99% of human rabies deaths [3]. Humans become infected by the rabies virus via the bite of a rabid animal. Prompt provision of post-exposure prophylaxis (PEP) is essential to prevent the development of disease in bite victims. However, in many rabies endemic countries, PEP is expensive and often unavailable [1]. Mass vaccination of domestic dogs has been shown to significantly reduce the incidence of rabies, both in human and dog populations [4], and several studies have demonstrated that targeting the source of rabies infection through dog vaccination is the most cost-effective approach to human rabies prevention [5,6,7].

Centralized pulsed vaccination is the standard delivery method for dog vaccination across much of Africa [8,9,10,11,12]. This method involves teams of vaccinators setting up temporary static-point clinics in a convenient central location that typically last for one day [13,14]. At least 70% of the dog population needs to be vaccinated in each annual campaign [15,16], to prevent herd immunity dropping below the critical threshold in the 12 month interval before the next campaign [16,17,18]. However, studies have shown that annual campaigns sometimes fail to attain this recommended coverage [4,19,20,21,22].

Herd immunity can decline by 30–50% per year in rabies endemic countries [19,23] due to high dog population turnover [11,17,19]. With the elimination of rabies dependent on continuously high vaccination coverage, alternative approaches to pulse vaccination may help to maintain herd immunity over time [24]. Thermotolerant rabies vaccines can be stored outside the traditional cold chain conditions (2 °C to 8 °C) for extended periods of time using controlled temperature chain systems [25], such as passive cooling devices [26,27,28,29]. The availability of thermotolerant vaccines, and affordable devices for storage of vaccines in settings with limited cold chain [30], open up avenues for the development of improved strategies for sustaining vaccination coverage across populations.

In this article, we report results from a feasibility study that was carried out to generate data to inform the design of a large-scale dog vaccination trial in northwest Tanzania. Specifically, the feasibility study was designed to: (a) investigate approaches to implement community-based mass dog vaccination; (b) compare three methods of assessing the vaccination status of individual dogs and the level of vaccination coverage in the community (at the ward level); and (c) assess travelling and waiting times at central point clinics and dog owner’s perceptions regarding the quality of services offered under three models of community-based dog vaccination delivery.

## 2. Materials and Methods

The aim of the study was to compare dog vaccination delivery approaches, specifically routine centralized pulsed vaccinations (CPV) normally led by a government team of vaccinators vs. novel decentralized continuous approaches to vaccination (DCV), which involve a Rabies Coordinator (RC) designated for this role by the district veterinary office (typically a livestock field officer) and a village-based worker, that we refer to as a One Health Champion (OHC). A potential advantage of a decentralized approach is that vaccines can be stored locally where they are needed rather than at the district veterinary office headquarters. Since decentralized continuous vaccination is novel, and the best methods for implementation were not clear, an important objective was therefore to compare the following alternative delivery strategies of this approach:

*Village-level continuous vaccination strategy (VLC):* RCs and OHCs worked together to deliver a static-point clinic in month 1 at a designated central point within each village. After the vaccination activities in month 1 were completed, the OHC continued to compile a list of dogs in their village that either missed the central point vaccination in month 1, were brought into the village or were born after month 1. These dogs were vaccinated when the RC returned to the village in months 3, 6, or 9. The RC together with the OHC were assigned to work together to decide whether the dogs requiring vaccination in months 3, 6 and 9 were to be brought to a central point for vaccination or whether these dogs would be better reached through a house-to-house approach. On demand vaccination services were also provided throughout the year, whereby dog owners called the RCs to come and vaccinate their dogs at their home.

*Sub-village**level continuous vaccination strategy (SVLC):* The RCs and OHC hosted a static-point clinic in month 1 at a designated central point within each sub-village of each village (there are typically 3 to 9 sub-villages per village in Tanzania). The subsequent vaccination activities were carried out as explained in the village-level strategy detailed above.

*Discretionary**continuous vaccination strategy (DC):* Prior to the beginning of the campaign, the RC, OHC and village leader held a consultative meeting and agreed on the vaccination strategy they considered to be the best for their village based on their local setting. That is, static-point clinics at either the village or the sub-village level, house-to-house or a mixture of the two strategies were agreed upon. As with the previous strategies, on demand vaccination was also to be provided throughout the year.

All vaccination approaches were provided free of charge.

### 2.1. Logistics and Dissemination of Equipment

Global Animal Health Tanzania purchased vaccination equipment (syringes, needles, certificates, smartphones, cool boxes, vaccine storage devices, etc.) and MSD Animal Health donated vaccines (Nobivac^®^ Rabies, MSD Animal Health, Boxmeer, The Netherlands). RCs were loaned smartphones for data collection and communication. Supplies were delivered to the district veterinary office in the quantities required, while the vaccine storage devices “Zeepots” [30] were sent directly to the wards where RCs were based.

### 2.2. Study Sites

This study was conducted between July 2019 and July 2020 in 12 wards of Butiama, Tarime and Rorya districts of the Mara region (34°–35° E, 1°30′–2°10′ S) in northwest Tanzania (Figure 1). Four wards were selected randomly from all wards in these districts that had a livestock field officer in each of these three districts. Wards in Tanzania typically comprise 4 villages but can have as many as 6, whereas the number of sub-villages within the village varies from 3 to 9. According to the 2012 national population and housing census, approximately 142,886 people live in the study area, i.e., in the 12 selected wards [31]. The dog population in these wards was estimated to be 23,814 dogs based on the human to dog ratio of 6:1 [14,32].

### 2.3. Dog Vaccination Campaigns

#### Centralized Pulsed Vaccination

One of the four selected wards in each district was randomly assigned to receive pulse vaccination, with all villages within this ward receiving the same vaccination delivery treatment. The district veterinary officer in each respective district managed the vaccination team, comprising three people, the vaccinator, certificate writer and data collector. The team held a one-day central point dog vaccination clinic in each village, after which these villages did not receive any further dog vaccinations for the remainder of the year. Prior to the campaign, communities were sensitized as previously described [20]. Briefly, dog owners were informed through posters placed in prominent places, mainly markets, schools and village notice boards, approximately 7 days before the campaign. One day before the vaccination campaign, a member of the vaccination team delivered announcements using megaphones to invite people to bring their dogs for vaccination the following day. The advertiser used a motorbike to reach different parts of the village. Dogs that were presented at the vaccination clinic were registered in a vaccination register, recording the owner’s name and the dog’s name, sex, and age. The same information were also entered into the Mission Rabies App [33]. Following registration, all dogs were inoculated subcutaneously with 1 mL of Nobivac Rabies vaccine, a microchip was then inserted subcutaneously into the scruff of the neck and a vaccination certificate was issued to the person who brought the dog to the central point.

### 2.4. Decentralised Continuous Vaccination

Under the DCV approach, a batch of vaccine as well as other vaccination equipment were stored and managed by the livestock field officer assigned to be the RC for the ward. The RC in collaboration with a community representative (the OHC) planned and executed vaccination activities throughout the year as needs arose. As with the CPV method, the district veterinary officer provided overall supervision of the vaccination.

### 2.5. Recruitment and Training

A local villager was recruited from each participating village to take responsibility as the OHC. Their appointment was performed by community leaders. To qualify for this position, the person had to meet specific criteria that included being influential and respected in the local community and having basic literacy. The RC and assigned OHC were then invited to attend a three-day training program aiming to provide specific skills and knowledge needed to conduct their assigned strategy for DCV. During the training the RCs were taught how to request and store vaccines in the Zeepot. In addition to the strategy specific training, the RCs and OHCs were taught other aspects of the vaccination programme including: (a) their roles and responsibilities; (b) planning and delivering vaccination campaigns according to their designated strategy; and (c) data collection and microchipping vaccinated dogs. Implementation manuals were provided to both RCs and OHCs for referral.

### 2.6. Pre-Vaccination Preparations

Within a week of the training, the RCs and OHCs developed vaccination planning timetables and shared these with local community leaders. The RCs submitted requests of their needs to the district veterinary officer based on the estimated dog population in their ward. In contrast to CPV, advertising was undertaken by OHCs who used loudspeakers and word-of-mouth to sensitize the community about the forthcoming dog vaccination campaign. Posters detailing the date and locations of the clinics were placed in the village notice board and in marketplaces. The RCs travelled by motorbike or public transport to the district veterinary office to collect vaccines in a cool box as well as other consumables required. Upon returning to their home base, RCs stored vaccines in their Zeepot, as previously described [30]. Each batch of vaccines that was issued was estimated to be sufficient to allow for six months of vaccination activities. To ensure dogs were vaccinated with high quality vaccine, any vaccines remaining at the end of the six-month period were returned to the district veterinary office and destroyed, and a new batch of vaccines was delivered. Once set, the OHC delivered another advertisement by loudspeaker the day before the vaccination campaign inviting people to bring their dogs and cats the following day. Each RC conducted the campaign as per the strategy assigned during the training.

### 2.7. Household Surveys

Surveys to assess vaccination coverage were conducted at two timepoints, from September to October 2019 and June to July 2020, respectively, corresponding to one month and 11 months after the initial vaccination campaigns. Different households were visited at each timepoint. The survey team was composed of the interviewer and the sub-village leader who provided guidance on the movements and introductions to households. The survey involved sampling in all villages in the twelve study wards. In each village three sub-villages were randomly selected and 10 households per sub-village sampled. With the help of the sub-village leader, a household (located at the far end of the sub-village) was chosen as the index household where the survey commenced. From this household the interviewer walked in a zigzag pattern, towards the center of the sub-village to the opposite side of the sub-village, sampling every fifth household. If no adult was present in the selected household or no dog was owned in the household, that household was skipped. Interviews continued until 10 households were completed in each sub-village. Prior to the administration of the questionnaire, permission was sought from the household head, or other household member of at least 18 years of age in the absence of the household head. Interviewers explained the study background to each respondent and obtained written informed consent to carry out the questionnaire. The questionnaire captured details regarding the quality of the dog vaccination services provided by the RCs and OHCs, the accessibility and availability of these services (e.g. how easy it was to travel to the vaccination point), the total number of dogs living in the household, and their vaccination status, according to whether either a microchip was found, a certificate was shown, and the owner statement (recall). During household visit at timepoint 1, it was noted that not all dogs belongings to the household were always available at home (dogs were roaming or accompanying livestock into the field) to assess their vaccination status. Thus, at the second time point, data were also collected on the number of dogs that were present at the time of the visit in addition to the total dogs belonging to the household) and therefore available for microchip checking. Reasons for not vaccinating dogs, and whether there were any unvaccinated dogs found at the household, were also recorded.

### 2.8. Statistical Analyses

The accuracy of assessing a dog’s vaccination status using certification and owner recall were assessed against the assumed gold standard of microchipping. Sensitivity and specificity were estimated with 95% confidence intervals calculated using the binom.test function in R. In this context, the sensitivity of the certification and owner recall tests were regarded as the ability to correctly identify dogs that have been vaccinated whereas the specificity referred to the ability of a test to correctly identify dogs that have not been vaccinated. Because test results were recorded at the household level and not for individual dogs, only single-dog households visited at timepoint 1 were included in this analysis.

Vaccination coverage at each of the two survey time points was estimated for each of the four vaccination strategies as the proportion of dogs with microchips among the surveyed dog population. Ideally the denominator in this coverage estimation would be the number of dogs in the household that were checked for microchips, but, since this information was not collected at the first time point, we used the total number of dogs living in surveyed households. Since this total number of dogs includes dogs that were roaming at the time of the survey and could not be checked for a microchip, the resulting coverage values are slight underestimates; unchecked dogs are effectively assumed to be unvaccinated. 

Assessment of coverage via microchip (when using as a denominator the number of dogs checked for microchips at the household; only available for the second survey time point) can be used as a gold standard from which to evaluate the accuracy of using vaccination certificates and owner recall to estimate coverage. This evaluation of coverage estimation methods was achieved by fitting generalized linear mixed models (GLMMs) with binomial error distributions to three different household-level response variables: (1) the number of microchipped dogs out of all dogs seen in surveyed households at time point two; (2) the number of vaccination certificates seen out of all dogs recorded as living at surveyed households surveyed at time point two; and (3) the number of dogs that owners reported to be vaccinated out of all dogs recorded as living at surveyed households surveyed at time point two. District and vaccination strategy were included as categorical fixed effects in all three GLMMs, along with random intercepts for ward, village, sub-village, and household (the observation level). The DHARMa package [34] was used to simulate residuals and check model assumptions were met. Estimates of the time point two vaccination coverage for each study ward were extracted from each model, with confidence intervals estimated from 1000 bootstrap samples using the bootMer function from the lme4 package [35].

Variations in duration of time travelling to and waiting at the central point clinics was modelled using a linear mixed model (LMM) fitted using REML, where log_e_ (duration) was the response, intervention and district were categorical fixed effects, and nested random intercepts were fitted at the ward, village and sub-village levels. The response was log-transformed to meet the assumptions of homoscedasticity and normality of residuals, which were checked visually by plotting the residuals against the fitted values. Differences between districts and intervention arms in mean of log_e_ duration of travel times and wait times at the central point clinics were tested using an F-test. Wald 95% confidence intervals for mean duration within each intervention arm were calculated in the lmerTest package [36] (“lmerTest”) using Satterthwaite’s approximation to the t-distribution degrees of freedom. Back-transformed mean durations of travel and wait times at the central point clinics and their 95% confidence limits from the log scale were adjusted for bias due to Jensen’s inequality [37]. Data analyses were performed using the statistical programming software, R version 3.5.3 [31].

## 3. Results

### 3.1. Vaccinations

In total 17,571 dogs were vaccinated in the study areas, as summarized in Table 1. Of these animals, 2654 were vaccinated through CPV; 5109 through DCV under the VLC strategy; 5172 through DCV under the SVLC strategy and 4636 through DVC under the DC strategy. Over the same period 2877 cats were vaccinated, with 178 vaccinated through CPV and 595, 939 and 1185 vaccinated from the three DCV strategies (Table 1).

During the main campaign, the vaccinators undertaking the CPV and the VLC vaccination strategies vaccinated the most dogs per day on average compared to those implementing the SVLC level and DC strategies (Table 2). For example, in Buswahili ward where VLC was being implemented, the number of dogs vaccinated per day by a vaccinator was 557 dogs in Batanga village, 442 dogs in Wegero village, 334 dogs in Buswahili village and 184 dogs in Kongoto. Subsequent campaigns in the DCV approach resulted in further dogs being vaccinated that would have been left unvaccinated if CPV was used. About 1642 more dogs were vaccinated in the follow-up campaigns under the VLC strategy (32.1% of all dogs vaccinated under VLC), while follow-up campaigns in the SVLC and DC strategies led to 1456 dogs (28.2% of all dogs vaccinated under SVLC) and 1139 dogs (24.6% of all dogs vaccinated under DC), respectively, being vaccinated (Figure 2). Of the 14,917 dogs vaccinated under the DCV approach, 14,196 (95%) were vaccinated using central point clinics held at village and sub-village levels, and only 2.7% and 2.1% of dogs were vaccinated using house-to-house and on-demand methods, respectively.

### 3.2. Accuracy of Dog Vaccination Status Assessment

The accuracy of certification and owner recall was assessed against the gold standard microchip using data from 598 single-dog households. The sensitivity and specificity of certification of dog vaccination status were 91.9% (95% CI 88.3–94.6) and 83.5% (95% CI 78.6–87.6), respectively. Owner recall of dog vaccination status had a sensitivity of 98.1% (95% CI 96.0–99.3) and specificity of 79.5% (95% CI 74.3–84.1).

### 3.3. Vaccination Coverage

The vaccination coverage achieved varied under the different delivery strategies (Figure 3). At time point 1 (1-month post-vaccination campaign), the overall vaccination coverage achieved using the DCV method was higher at 64.1% (95% CI 62.1–66.0) than under the CPV approach at 35.9% (95% CI 32.6–39.5). Similarly, at time point 2 the DCV method had a higher level of coverage of 60.7% (95% CI 58.5–62.8) compared to CPV with 32.1% (95% CI 28.8–35.6). Between the two time points, the coverage dropped by 3.8% and 3.4% in the CPV and DCV methods, respectively. 

Estimation of vaccination coverage for each ward using the GLMM for owner recall (Figure 4; triangles) overestimates coverage (relative to estimates from the GLMM for the gold standard approach using microchips) by an average of 2.5 percentage points, though this difference between the two estimation methods varies between wards and strategies. Overestimation by the owner recall approach is most substantial under the CPV strategy and is less evident under the DCV strategies, where underestimation sometimes occurs. Coverage estimates from the GLMM for certificates (Figure 4; squares), however, underestimates coverage on average by 3.6 percentage points, with underestimation being most substantial for the DCV strategies (certificates tended to overestimate vaccination under CPV). We note though that for every ward, the 95% confidence intervals for vaccination coverage estimated by owner recall and by certificates always overlap the 95% confidence interval for coverage estimated by microchips; suggesting that differences between the methods are not significant. 

### 3.4. Vaccination Services

Over 85% of all respondents across the strategies found it easy or very easy to travel from their household to the vaccination point (Figure 5). Table 3 reports respondents perceptions about the dog vaccination services provided that resulted from the household survey. The proportion of respondents who were aware of the presence of the RC in their ward was lower in areas vaccinated at the sub-village-level (67%) compared to village-level (83%). More than 85% of people who were aware of the RC knew how to contact them if they needed the service. About half of the respondents in DC strategy were aware there was an OHC in their village, whilst the majority of respondents (>80%) in the other DCV strategies were aware of the presence of the OHC and >90% of them knew how to contact the OHC. Overall, villagers were satisfied (>84%) with the services they received from the RC and OHC. 

### 3.5. Reasons for Not Vaccinating Dogs

Table 4 summarizes the reasons given for not vaccinating dogs. There was a significant difference between vaccination strategies in the reason reported by respondents for not vaccinating their dogs. In those villages targeted with the CPV approach 22% of respondents cited lack of awareness about the campaign as the main reason for not vaccinating their dogs, whereas only 7% of the respondents in the DCV sub-strategies indicated lack of information regarding the vaccination campaign as the reason for not vaccinating dogs. 

The vast majority of respondents with unvaccinated dogs in the areas under the DCV approach reported that their dogs were not vaccinated because they were acquired or born after the campaign was conducted (41.51%, Table 4). In contrast, most respondents with unvaccinated dogs in the areas under the CPV approach reported that they didn’t hear about the campaign or the owner was unavailable to take them to be vaccinated. Although the DCV was supposed to be available throughout the year, the results suggest that there were fewer vaccination campaigns conducted than originally planned in the implementation manual. In both approaches, distance to the central point clinic was rarely mentioned as a reason for not vaccinating the dogs (0.5% and 0.3% in the pulse and continuous strategies, respectively). Similarly, dogs running away comprised a large proportion of dogs that were not vaccinated in both approaches.

Differences between districts and intervention arms in mean of log_e_ duration of travel times and wait times at the central point clinics are shown in Table 5. There was not a statistically significant difference in travel and wait time at the central point clinic between districts and interventions (F(2, 3) = 0.6, *p* = 0.56) and (F(2, 3) = 3.4, *p* = 0.09) respectively. Thus, district was dropped from the model. 

Table 6 summarizes the time that dog owners reported traveling from their house to the central point, and the time spent at the central point clinic waiting for their dog(s) to be vaccinated. There was a significant difference in mean travel time to the central point vaccination clinics (df = 3, F = 28.9, *p <* 0.0001) between the strategies. There was a significant difference in mean travel time to the central point clinics between the SVLC and the VLC (*p* < 0.0001), whereby people in the VLC spent 11 min more to locate the central point clinic compared to people under SVLC strategy. Also, people under DC strategy spent 8 min less to locate the central point clinic compared to people under VLC (*p* < 0.0001). Similarly, there was a significant difference in mean wait time at the central point vaccination clinics (df = 3, F = 13.1, *p <* 0.0001) between the strategies. There was a significant difference between the SVLC strategy and the CPV (*p <* 0.0001) with people receiving CPV strategy waiting on average 25 min more than those under the SVLC strategy. There was also a significant difference between DC and CPV (*p <* 0.0001) with people under CPV strategy waiting on average 23 min more than those under the DC strategy.

## 4. Discussion

This study describes the development and implementation of a novel approach to the delivery of mass dog vaccination, taking advantage of a thermotolerant rabies vaccine [25] and a locally made and tested passive cooling device [30]. Although the trial was not powered to allow statistical comparison between the two delivery strategies, the application of DCV attained higher initial vaccination coverage than the standard method CPV and sustained greater coverage over time following measurement at 11 months.

Our findings are consistent with previous studies in rural Sub-Saharan Africa, which show that few pulsed vaccination campaigns manage to reach the recommended annual coverage of 70% [4,8,20,21,38,39]. Hampson et al. [17] reported that vaccinating 60% of the dog population through CPV conducted on the annual basis should be sufficient to prevent coverage declines to below the critical threshold, with only very few and small outbreaks ever observed in areas with coverage of 60% or more. This is supported by a previous study in northwest Tanzania that attained a coverage of 64%, and resulted in significant reductions of the incidence of rabies in both humans and dogs [4]. Coverage of 60–70% is recommended for campaigns conducted annually to avoid coverage falling below the critical threshold coverage of 20–40% in the interval between vaccination campaigns. In this study, overall coverage in the DCV declined from 64.1% to 60.7% at months 1 and 11, respectively, which is well above the critical threshold [16] needed to prevent the transmission of rabies virus [1,2,3].

Vaccination coverage is frequently assessed at only one timepoint, usually within a few days of campaigns; however, herd immunity wanes over time due to population turnover [19,40,41]. Thus, high birth and death rates may result in declines in coverage even if the campaign attains high initial coverage [9,12,18]. One way to maintain effective vaccination coverage is to ensure vaccination services are provided free of charge and continuously as the need arises. Continuous vaccination as applied in this study apparently helped to stabilize the coverage at relatively constant levels throughout the study period, with 4237 dogs (28.40%) vaccinated via follow-up campaigns conducted under the DCV approach.

House-to-house vaccine delivery or the combination of central point vaccination clinics and house-to-house delivery strategies have proven to be effective for achieving 70% coverage across a variety of dog populations [20,42,43]. For example, Morters et al. [44] reported a coverage of up to 82% through house-to-house visits and Gibson et al. reported a vaccination coverage of 79.3% which was the result of static point and house-to-house vaccinations [42]. In our study, only 2.7% and 2.1% dogs were vaccinated using house-to-house and on-demand, respectively, with 95% of dogs vaccinated at central point clinics. There are several possible reasons to explain this from the perspective of supply (service providers) and demand (community). The household surveys revealed that most respondents were not aware of the availability of house-to-house and on-demand services, while some owners feared that calling a RC to vaccinate their dogs would incur a charge. During follow-up campaigns the RCs and OHCs were instructed to vaccinate dogs using any method they considered the best; however, in practice they probably preferred central point clinics as they are logistically more simple and less labor intensive compared to house-to-house and on-demand [22,45]. Moreover, as government employees, RCs have other administrative tasks that might compete with their time to conduct intensive house-to-house and on-demand campaigns. 

Data from this study suggested that owner recall tended to overestimate coverage at the ward level, particularly under CPV, while vaccination certificates tended to underestimate coverage. These results concurred with our data on accuracy of assessment at the individual dog level, with owner recall being more sensitive than certification, while both methods had similar specificities. Overestimation as a result of owner recall is not unexpected given dog owners are legally required to have their dogs vaccinated and as such there is an incentive to report that their dogs have been vaccinated. Underestimation by certificates is likely due to poor storage or loss of certificates [8,12,14]. There were also some cases of overestimation by certificates though, possibly due to presentation of invalid or expired certificates, or certificates being shared between dogs [45]. However, the differences between coverage estimates via the two alternative methods and the gold standard microchipping method were non-significant, suggesting that certificates and owner recall may both be reliably used as cheap alternative methods to assess coverage.

Lack of awareness regarding vaccination campaigns appeared to be a major obstacle for people to vaccinate their dogs. Despite the efforts made to sensitize the community about vaccination campaigns, 22.6% of respondents in the areas receiving CPV who did not vaccinate their dogs reported being unaware of the campaign as the reason for not participating. Lack of awareness of vaccination campaigns has been mentioned in other studies as a leading reason for failure to attend vaccination clinics [14,45,46,47,48]. In contrast, only 1.8% of people who received DCV reported lack of awareness as a reason for not vaccinating dogs. Integrating OHCs into the DCV approach could underlie the improved awareness of the campaigns in this method as, among other duties, OHCs played a role in sensitizing the community about rabies campaigns and communicating vaccination schedules.

Introduction of new dogs from outside the study area and the birth of new puppies was mentioned (41.5%) as the main reasons for dogs not being vaccinated in the DCV method. Although the vaccination campaigns were supposed to be continuous, the household surveys indicated that the RCs performed few subsequent campaigns after the major campaign in month one.

In contrast to other studies which have shown that coverage declines with increased distance [20] and the likelihood of people attending central point vaccinations decreases with increasing distance [45,49,50,51], very few people in this study (0.5% in CPV and 0.3% in DCV) perceived distance as an obstacle to their participation (Table 4 and Figure 5). This finding is similar to a study in Uganda which also showed that people were willing to travel large distances to reach vaccination points [52]. Another study in Zambia reported that dog owners were willing to travel up to 5 km [8], whereas a study in Tanzania found walking a long way to vaccination clinics was not given as reason for non-participation in the vaccination campaign [4] and in Chad owners did not perceive distance as a major obstacle [13]. One study that investigated the barriers to attendance at central point vaccination found that people who did not attend the vaccination campaigns quoted being unaware of the vaccination campaign as the main reason for non-attendance (27%). Further exploration indicated that these people were located far from the central point out of reach of advertisements [45]. This suggests that being unaware of the campaign rather than distance could play a major role in influencing central point attendance. 

Assessing service delivery can provide understanding of how the quality of the services offered matches owners’ perceptions and expectations. Experience from other studies shows that feedback provided through such dialogue helps to improve service and maintain dog owners’ interest in the programme [53]. The results from the assessment of service provided by RCs and OHCs revealed that the majority of respondents (84.7% in DC, 91.2% in SVLC and 91.3% in DC) were satisfied. Their satisfaction was highlighted on items like waiting time at the vaccination point, availability of vaccination services throughout the year, affordability of the services i.e. offered free of charge, provision of more flexible delivery strategies such as house-to-house, on demand and sub-village level campaigns, as well as RCs and OHCs adherence to the vaccination schedule and time. In addition, integration of community members into the program improved participation and may have helped bring a greater sense of ownership which is vital for the success of such a programme. 

Dog owners under the SVLC strategy spent less time to reach vaccination points and less time (<30 min) to complete registration and vaccination at the central point (for the DC strategy also). This makes sense, given that VLC campaigns involves the entire village attending in a single day. During this study, some central point clinics at VLC experienced large turnouts. For example, in Baranga village in Buswahili ward, 557 dogs were vaccinated at one central point clinic in a single day. Registration, certificate writing and vaccination of such large numbers of dogs likely contributed to longer waiting times. However, vaccinators at the VLC also vaccinated more animals per day compared to vaccinators under the SVLC and DC strategies (Table 2). 

We did not assess or compare the cost-effectiveness of these vaccine delivery strategies and cost-effectiveness is an important consideration for large-scale interventions that should be examined in future. All the delivery strategies were effectively government-led interventions utilizing existing animal health delivery infrastructure and government employees, and therefore do provide a model for scaling up mass dog vaccination to other parts of the country. Other studies of animal health or public health interventions suggest that decentralized government-directed interventions are more cost-effective [54] and sustainable [55,56]. Overall, this feasibility study provided clear evidence to support the use of a decentralized continuous approach to dog vaccination which will be fully evaluated through a randomized controlled trial implemented across the entire Mara region.

## 5. Conclusions

This study is the first to report the implementation of mass dog vaccination through a continuous vaccination approach using thermotolerant rabies vaccines stored locally in passive cooling devices (Zeepots). It also provides an estimate of the vaccination coverages achieved and maintained at critical timepoints post-vaccination. Our results suggest that a DCV approach is feasible to implement and can achieve and maintain sufficiently high coverage of the dog population. The deployment of the Zeepot and engagement of One health champions appeared to improve access to vaccines as well as community awareness and participation in the campaign. Combining various vaccination strategies in a continuous delivery approach can support the goal of maintaining coverage at sufficiently high levels throughout the yearly cycle. We hope that the results from this study will support the design and implementation of more effective dog vaccination campaigns to achieve the goal of dog-mediated human rabies elimination by 2030.

## Figures and Tables

**Figure 1 viruses-14-00830-f001:**
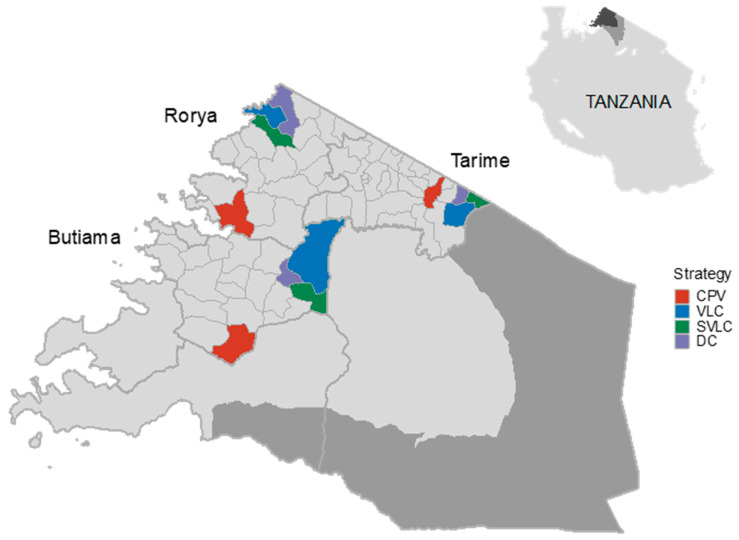
Map of the Mara region in Tanzania showing study wards and the assigned vaccination delivery approaches. Approaches comprised centralized pulsed vaccination (CPV), and three strategies of decentralized vaccination, specifically village-level continuous (VLC), sub-village level continuous (SVLC), and discretionary continuous (DC).

**Figure 2 viruses-14-00830-f002:**
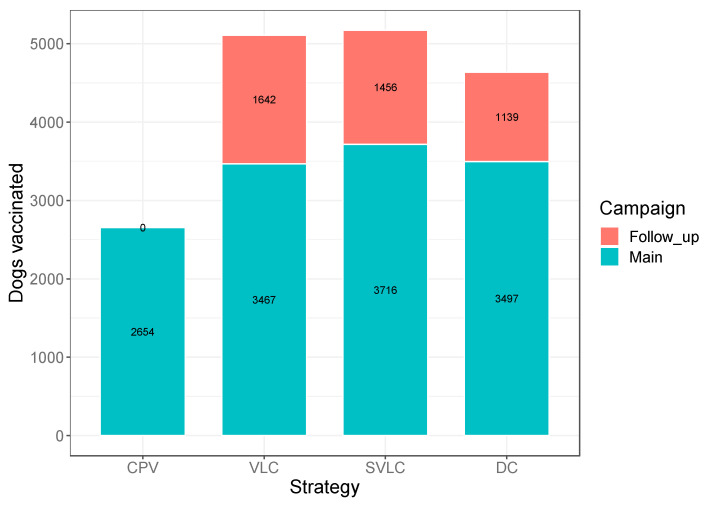
Dogs vaccinated in each strategy in the main and follow-up campaigns. CPV = centralized pulsed vaccination, VLC = village-level continuous decentralized vaccination, SVLC = sub-village level continuous decentralized vaccination, DC = discretionary continuous decentralized vaccination.

**Figure 3 viruses-14-00830-f003:**
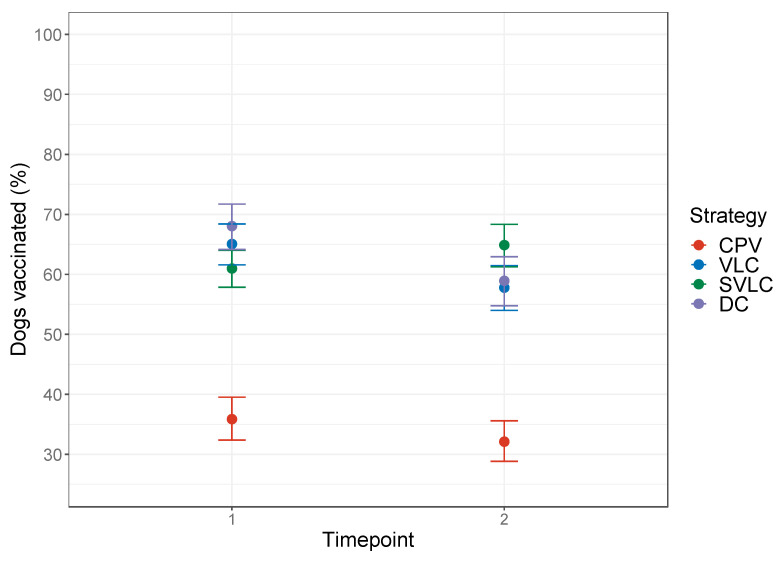
Vaccination coverage attained in each strategy at month 1 and month 11 post vaccination. Coverage was estimated from the number of dogs with microchips out of all dogs recorded as living at surveyed households. CPV = centralized pulsed vaccination, VLC = village-level continuous decentralized vaccination, SVLC = sub-village level continuous decentralized vaccination, DC = discretionary continuous decentralized vaccination.

**Figure 4 viruses-14-00830-f004:**
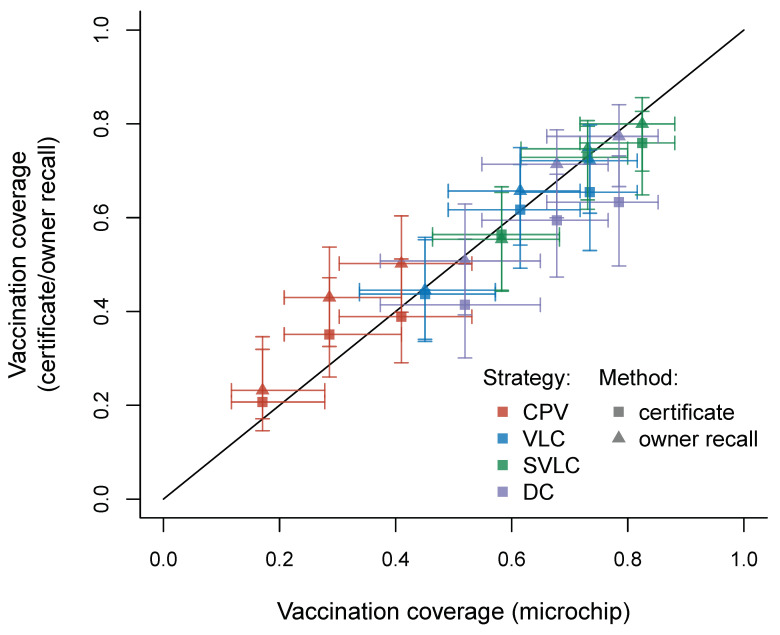
Coverage estimates according to different methods of measuring coverage. Generalized linear mixed model (GLMM) estimates in each of the twelve study districts under each of the three coverage estimation methods at 11 months after the main vaccination campaign. 95% confidence intervals estimated using 1000 bootstrap samples from the fitted GLMMs are included for the estimates obtained using certificates and owner recall (vertical intervals) and using the gold standard microchip method (horizontal intervals). CPV = centralized pulsed vaccination, VLC = village-level continuous decentralized vaccination, SVLC = sub-village level continuous decentralized vaccination, DC = discretionary continuous decentralized vaccination.

**Figure 5 viruses-14-00830-f005:**
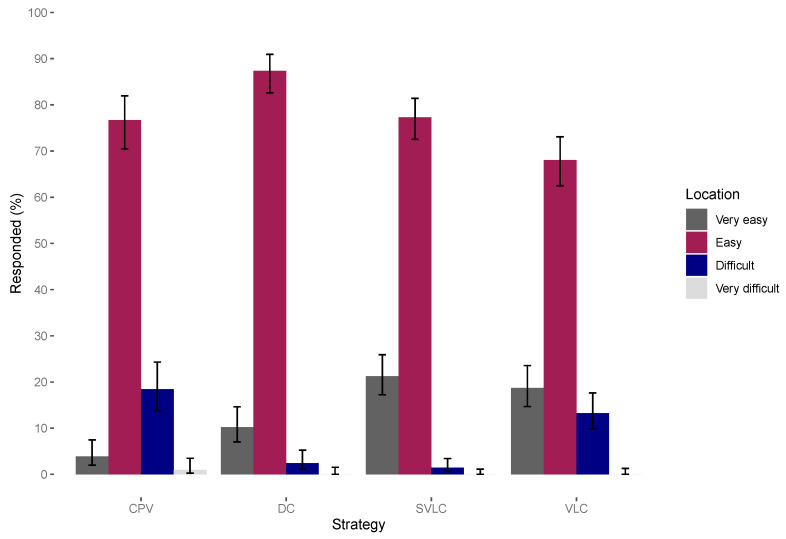
Respondent’s perceptions regarding the location of the household relative to the central point clinic. CPV = centralized pulsed vaccination, VLC = village-level continuous decentralized vaccination, SVLC = sub-village level continuous decentralized vaccination, DC = discretionary continuous decentralized vaccination.

**Table 1 viruses-14-00830-t001:** Numbers of dogs vaccinated across the different strategies.

Sn	District	Ward	Strategy	Dogs
1	Butiama	Masaba	CPV	524
2	Butiama	Buswahili	VLC	2198
3	Butiama	Sirorisimba	SVLC	2403
4	Butiama	Nyamimange	DC	1594
5	Rorya	Komuge	CPV	1326
6	Rorya	Tai	VLC	1418
7	Rorya	Mkoma	SVLC	1280
8	Rorya	Bukura	DC	1606
9	Tarime	Nyamwaga	CPV	804
10	Tarime	Gorong’a	VLC	1493
11	Tarime	Nyanungu	SVLC	1489
12	Tarime	Itiryo	DC	1436
	Total			17,571

**Table 2 viruses-14-00830-t002:** Average number of dogs and cats vaccinated per day during the main campaign in each strategy.

Strategy	Dogs
CPV	223
VLC	275
SVLC	72
DC	71

**Table 3 viruses-14-00830-t003:** Respondent perceptions regarding the dog vaccination services under the three sub-strategies of the DCV.

Variable	VLC	SVLC	DC
Is the respondent aware there is a RC in their ward?	301/361 (83.4%)	233/348 (67.0%)	237/296 (80.1%)
If yes, do they know how to contact the RC?	256/301 (85.1%)	200/233 (85.8%)	235/237 (99.2%)
Is the respondent aware there is a OHC in their village?	330/361 (91.4%)	315/388 (81.2%)	150/296 (50.7%)
If yes, do they know how to contact the OHC?	303/330 (91.8%)	302/315 (95.9%)	144/153 (94.1%)
Are you satisfied with the vaccination services provided by the RC and OHC in your area?	324/355 (91.3%)	354/388 (91.2%)	249/294 (84.7%)

Notes: NA means non-available. CPV = centralized pulsed vaccination, VLC = village-level continuous decentralized vaccination, SVLC = sub-village level continuous decentralized vaccination, DC = discretionary continuous decentralized vaccination.

**Table 4 viruses-14-00830-t004:** Reasons for not vaccinating dogs under centralized pulsed vaccination and decentralized continuous vaccination strategies.

Reason	CPV (*n*)	%	DCV (*n*)	%
1. Dog had puppies	10	5.1	4	1.0
2. Dog sick	2	1.0	3	0.8
3. Dog aggressive	9	4.6	13	3.4
4. Acquired/born after vaccination	28	14.4	159	41.5
5. Didn’t hear about the campaign	44	22.6	7	1.8
6. Dog difficult to restrain	14	7.2	48	12.5
7. Distance to vaccination point too long	1	0.5	1	0.3
8. Dog not at home	7	3.6	23	6.0
9. Owner sick	7	3.6	3	0.8
10. Dog too young	2	1.0	12	3.1
11. Ran away	28	14.4	84	21.9
12. Owner unavailable	36	18.5	19	5.0
13. Not enough time	6	3.1	0	0.0
14. Mistrust of vaccination campaigns	1	0.5	0	0.0
15. Vaccine finished	0	0.0	3	0.8
16. Too many dogs at home	0	0.0	4	1.0
Total	195	100.0	383	100.0

Note: CPV = centralized pulsed vaccination and DCV = decentralized continuous vaccination.

**Table 5 viruses-14-00830-t005:** Results of F tests on the fixed effects using Satterthwaite approximation in wait and travel time to the central point.

	Fixed Effect	Sum of Squares	Mean Squares	Df (num, den.)	F Value	*p* Value
Travel time	Intervention	1.6	0.5	3, 6.1	1.3	0.37
	District	0.5	0.3	2, 6.2	0.6	0.56
Wait time	Intervention	3.1	1.0	3, 6.1	1.9	0.22
	District	3.7	1.9	2, 6.1	3.4	0.09

**Table 6 viruses-14-00830-t006:** Average time spent by dog owners at the central point and time taken to reach the central point.

Strategy	Mean Travel Time (min)	95% CI	Mean Wait Time (min)	95% CI
CPV	22.4	14.8–33.9	58.5	23.7–144.0
VLC	28.1	18.8–42.2	81.3	33.2–199.5
SVLC	17.3	11.5–25.9	34.0	13.8–83.3
DC	20.2	13.4–30.6	35.3	14.3–87.3

## Data Availability

Deidentified data and code to reproduce the analyses are available from Zenodo: DOI: 10.5281/zenodo.6463278.

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
