# Peer review of "Development of Dog Vaccination Strategies to Maintain Herd Immunity against Rabies"

_viruses, 2022, doi:10.3390/v14040830_

Round 1

Reviewer 1 Report

This publication is a contribution to the global strategic plan to eliminate human deaths from  dog -mediated rabies by 2030 report the implementation of mass dog parenteral vaccination through a continuous vaccination using rabies vaccin stored locally in passive cooling devices. The results obtained indicate that the decentralised method may give higher levels of immunisation in the dog population than the centralised method. 

However, as the authors emphasise, the success of such an action is largely dependent upon information and communication to the community. We do not know the result of a centralised campaign with a better organised information action.

Suggestion for authors:

  • the title of the paper is Development of dog vaccination strategies...not dogs and cats so  it is therefore reasonable to remove the data concerning cats from tables 1 and 2. Of course, the discussion could include information that an additional positive effect of the action was the vaccination of a number of cats, which increases the effect of the vaccination.
  • Abbreviations for the different methods of organising the vaccination are given in the materials and methods. It would be much easier to read the text if these abbreviations would be used throughout the whole text e.g. line 281, 282298, 299, 303 etc
  • table 3 and 4 please aadjust the text in the first column to the left
  • table 5 please specify the time unit
  • Conclusuion - can you explain the term thermotolerant rabies vaccine- is this a special rabies vaccine more resist to temperature

Author Response

Dear reviewer,

I really appreciate your time reviewing this work and for your positive feedback. Please see below our responses

Point 1: the title of the paper is Development of dog vaccination strategies...not dogs and cats so  it is therefore reasonable to remove the data concerning cats from tables 1 and 2. Of course, the discussion could include information that an additional positive effect of the action was the vaccination of a number of cats, which increases the effect of the vaccination.

Response 1: cat data has been removed from table 1 and 2

Point 2: Abbreviations for the different methods of organising the vaccination are given in the materials and methods. It would be much easier to read the text if these abbreviations would be used throughout the whole text e.g. line 281, 282298, 299, 303 etc

Response 2: Corrections have been made. We have used the abbreviations throughout the text as suggested

Point 3: table 3 and 4 please aadjust the text in the first column to the left

Response 3: Text has been formatted i.e. aligned to the left  

Point 4: table 5 please specify the time unit

Response 4: The unit of time (min) has been added in the table 5

Point 5: Conclusuion - can you explain the term thermotolerant rabies vaccine- is this a special rabies vaccine more resist to temperature

Response 5: we have described the meaning of thermotolerant vaccines lines 59-61

Reviewer 2 Report

Nice publication that highlights the need for adapted strategies for rabies vaccination in dogs.
Minor spelling mistake on line 439: 'Lock' in stead of "lack"

Author Response

Point 1: Minor spelling mistake on line 439: 'Lock' instead of "lack"

Response 1: Thank you for the positive feedback. The spelling error has been fixed on line 602

Reviewer 3 Report

Reviewer’s comments

Title: Development of dog vaccination strategies to maintain herd immunity against rabies

Journal: Viruses

The research work entitled “Development of dog vaccination strategies to maintain herd immunity against rabies” presents on the demonstrate an effective approach to maintain and prolong herd immunity against rabies. This work has made forward-looking progress in the design of a large-scale dog vaccination trial. The whole article is excellent in logical structure, description of existing problems, data presentation, and argument advocacy. There are only a few minor formatting flaws. I suggested this work may be “minor revision” for publication in “Viruses”. Specific comments and general comments are given below:

Specific comments

  1. Line 351-356, the information here is suggested to attach the source.
  2. The column Pulse in Table 3 has some information blank. To increase readability, I suggest that even non-available (NA) should be filled in, and the meaning of NA should be noted below the table.
  3. I recommend authors to standardize the format of journal names and DOI numbers in References to an abbreviated format. For example: Bulletin of the World Health Organization --> World Health Organ; Preventive Veterinary Medicine --> Prev. Vet. Med; Proceedings of the National Academy of Sciences --> Proc. Natl. Acad. Sci. U.S.A. On the other hand, the DOI numbers are also recommended to be provided consistently.

General comments

  1. Line 86, the village in months 3, 6 or 9. --> the village in months 3, 6, or 9. (missing comma).
  2. …provided free of charge --> … provided free of charge. (missing dot).
  3. Line 371, 373, and 375, p < 0.0001 --> p < 0.0001 (change to italic).
  4. Line 346, 347, 348, >XX% à > XX% (mathematical symbols and numbers are separated by white space).
  5. Line 410 et al --> et al. (missing dot).
  6. Line 475 (<30 min) --> (< 30 min) (mathematical symbols and numbers are separated by white space).

Author Response

Dear reviewer

Thank you for the positive feedback, this is higgly appreicaited. Please see the responses below

Specific comments

Point 1: Line 351-356, the information here is suggested to attach the source.

Response: Source has been added

Point 2: The column Pulse in Table 3 has some information blank. To increase readability, I suggest that even non-available (NA) should be filled in, and the meaning of NA should be noted below the table.

Response 2: Non-available (NA) has been filled in the empty spaces and meaning given below the table

Point 3: I recommend authors to standardize the format of journal names and DOI numbers in References to an abbreviated format. For example: Bulletin of the World Health Organization --> World Health Organ; Preventive Veterinary Medicine --> Prev. Vet. Med; Proceedings of the National Academy of Sciences --> Proc. Natl. Acad. Sci. U.S.A. On the other hand, the DOI numbers are also recommended to be provided consistently.

Response 3: Journal names have been standardized “abbreviated” and missing doi added

General comments

Point 1: Line 86, the village in months 3, 6 or 9.  the village in months 3, 6, or 9. (missing comma).

Response 1: comma has been added

Point 2: …provided free of charge --> … provided free of charge. (missing dot).

Response 2: missing dot has been added

Point 3: Line 371, 373, and 375, p < 0.0001 --> p < 0.0001 (change to italic).

Response 3: p values have been changed to italic

Point 4: Line 346, 347, 348, >XX% à > XX% (mathematical symbols and numbers are separated by white space).

Response 4: this has been fixed

Point 5: Line 410 et al --> et al. (missing dot).

Response 5: missing dot added

Point 6: Line 475 (<30 min) --> (< 30 min) (mathematical symbols and numbers are separated by white space).

Response 6: this has been fixed